# Dynamic transition of current-driven single-skyrmion motion in a room-temperature chiral-lattice magnet

Licong Peng [1✉], Kosuke Karube [1], Yasujiro Taguchi [1], Naoto Nagaosa [1,2], Yoshinori Tokura [1,2,3] & Xiuzhen Yu [1✉]

Driving and controlling single-skyrmion motion promises skyrmion-based spintronic applications. Recently progress has been made in moving skyrmionic bubbles in thin-film heterostructures and low-temperature chiral skyrmions in the FeGe helimagnet by electric current. Here, we report the motion tracking and control of a single skyrmion at room temperature in the chiral-lattice magnet $Co_9Zn_9Mn_2$ using nanosecond current pulses. We have directly observed that the skyrmion Hall motion reverses its direction upon the reversal of skyrmion topological number using Lorentz transmission electron microscopy. Systematic measurements of the single-skyrmion trace as a function of electric current reveal a dynamic transition from the static pinned state to the linear flow motion via a creep event, in agreement with the theoretical prediction. We have clarified the role of skyrmion pinning and evaluated the intrinsic skyrmion Hall angle and the skyrmion velocity in the course of the dynamic transition. Our results pave a way to skyrmion applications in spintronic devices.

[1] RIKEN Center for Emergent Matter Science (CEMS), Wako, Japan. [2] Department of Applied Physics, University of Tokyo, Bunkyo-ku, Japan. [3] Tokyo College, University of Tokyo, Bunkyo-ku, Japan. ✉email: licong.peng@riken.jp; yu_x@riken.jp

M agnetic skyrmions with vortex-like spin textures can be moved by ultralow current density and hence are promising candidates for information carriers in energy-efficient spintronic devices[1,2]. Skyrmions possess topological stability described by the integer character of their topological number $N_{sk}$,[3]

$$N_{sk} = \frac{1}{4\pi} \iint \boldsymbol{n} \bullet \left( \frac{\partial \boldsymbol{n}}{\partial x} \times \frac{\partial \boldsymbol{n}}{\partial y} \right) dx dy \qquad (1)$$

where $\boldsymbol{n} = \frac{\boldsymbol{M}}{|\boldsymbol{M}|}$, and $\boldsymbol{M}$ is the magnetization. This topological characteristic imparts skyrmions with particle-like properties, extraordinary metastability[4,5], and emergent electromagnetic phenomena[1–3]. The flexible shape deformation of skyrmions via the topological protection allows skyrmions to avoid impurities during the current-driven motions, in contrast to domain walls and helices[6–8]. Under electric current excitation, the skyrmion motion is driven by the spin-transfer torque (STT). In a system containing impurities, it can be described by the Thiele equation[6,9,10]

$$\boldsymbol{G} \times (\boldsymbol{v}_s - \boldsymbol{v}_d) + \boldsymbol{\mathcal{D}}(\beta \boldsymbol{v}_s - \alpha \boldsymbol{v}_d) + \boldsymbol{F}_{pin} = 0 \qquad (2)$$

where $v_d = \sqrt{v_x^2 + v_y^2}$ is the skyrmion drift velocity and $v_s$ is the velocity of the conduction electrons. The first term describes the Magnus force represented by the Magnus vector $\boldsymbol{G} = (0, 0, 4\pi N_{sk})$. The second term corresponds to the dissipative force related to the tensor $\boldsymbol{\mathcal{D}}$, where $\alpha$ is the Gilbert damping factor and $\beta$ is the nonadiabatic coefficient. The third term is the pinning force $\boldsymbol{F}_{pin}$ arising from impurities. The onset of skyrmion movement is determined by the competition between the underlying pinning sites within the materials and the driving force; that is, skyrmions are pinned by defects and substantially are moved by an external force when it exceeds a certain threshold. Hence the skyrmion motion exhibits a dynamic transition with increasing the driving current, i.e., from the static pinned state to the flow motion by way of a creep motion, as suggested by numerical simulations[11,12].

The dynamic transition of Néel-type skyrmionic bubbles with micrometer size, which is caused by the spin–orbit torque (SOT), has been demonstrated in thin-film heterostructures with interfacial Dzyaloshinskii–Moriya interaction (DMI)[13]. Driving skyrmionic bubbles requires a high current density of ~$10^{10}$–$10^{12}$ A m$^{-2}$ to overcome the randomly distributed pinning sites in synthetic multilayered films prepared by magnetron sputtering techniques[13–17]. In contrast, the nanoscale skyrmion dynamics driven by the STT in chiral-lattice magnets is attractive because of the ultralow onset-current density (~$10^6$ A m$^{-2}$) to move skyrmions[1,2]. Previous investigations have focused on the nucleation and motion of skyrmion clusters[18–20] since a single skyrmion is difficult to isolate in the thermodynamic equilibrium phase in chiral-lattice magnets. The first demonstration of a single-skyrmion torque motion has been reported in the FeGe helimagnet at 120 K, well below room temperature (RT)[20], whereas the control of single-skyrmion motion at RT is valuable for future applications in energy-efficient spintronic devices. However, a direct experimental demonstration of the dynamic transition of the single-skyrmion motion at RT in chiral magnets remains elusive.

Among various chiral-lattice magnets[19,21–25], Co$_9$Zn$_9$Mn$_2$ that can host RT skyrmions is a good target material for manipulating skyrmions with electric current. Co$_9$Zn$_9$Mn$_2$ has a noncentrosymmetric-cubic structure with the space group $P4_132$ or $P4_332$ (Fig. 1a). The transition temperature ($T_C$) from the paramagnetic state to the spin-spiral ordered state is ~396 K[24], i.e., well above RT, which is essential for avoiding skyrmion annihilation in applications. The equilibrium skyrmion lattice (SkL) occupies a narrow region in the temperature-magnetic field ($T$-$B$) phase diagram, just below $T_C$.[24] Meanwhile, a conical state,

which is thermodynamically stable at RT, provides a broad magnetization-polarized background for manipulating isolated skyrmions. Therefore, we choose Co$_9$Zn$_9$Mn$_2$ to demonstrate the current-driven single-skyrmion motion at RT.

In this work, we have created a single metastable skyrmion at RT in Co$_9$Zn$_9$Mn$_2$. We have directly tracked the single-skyrmion translational and Hall motions induced by nanosecond current pulses. By reversing the pulsed current direction, we could control the trace of the isolated skyrmion. By flipping the magnetic field, thus generating the skyrmion with an opposite sign of a topological number, we demonstrate the opposite Hall motion of the skyrmion transverse to the electric current direction. We also report the current-induced dynamic transition of the single-skyrmion motion from the pinned state to the flow motion via the creep motion, revealing the influence of pinning sites on the skyrmion dynamics.

## Results

**Creation of isolated skyrmion in Co$_9$Zn$_9$Mn$_2$-based microdevice.** We study the single-skyrmion dynamics in a microdevice composed of a (001) Co$_9$Zn$_9$Mn$_2$ thin plate (see Fig. 1b and Supplementary Fig. 1). The application of a magnetic field will only generate a conical state, while the SkL is suppressed at RT[24]. Therefore we use electric current pulses to create the metastable SkL in the thermodynamically stable conical phase at RT (Supplementary Fig. 2). A single skyrmion is then isolated (Fig. 1c) from the metastable SkL by controlled sweeps of magnetic field (see details in Supplementary Fig. 2). The metastable single skyrmion generated at −80 mT exhibits a small size of ~100 nm. The negative value of the applied magnetic field means that the field is directed along −$z$-direction, i.e., from the top to the bottom of the sample, which leads to the magnetizations pointing upwards at the skyrmion center and downwards at the periphery (Fig. 1d). Hence, the topological number $N_{sk}$ of the skyrmion is +1 according to its definition in Eq. (1). The magnetic induction field map of the skyrmion (Fig. 1c) indicates a clockwise helicity for the skyrmion, which is encoded by the hue-saturation-lightness color wheel. The surrounding background of the skyrmion displaying dark contrast encodes the polarized magnetization, i.e., the uniform magnetization of a conical state at a small magnetic field of −80 mT. The conical background isolates the skyrmion, which allows avoiding complicated interaction among skyrmions[13,26]. Upon excitation with a pulsed electric current ($j$) from right to left (as marked by the blue arrow in Fig. 1d), the skyrmion is expected to show a translational motion in the antiparallel direction to the current, and the Hall motion transverse to the current direction (schematically drawn in Fig. 1d).

**Current-driven single-skyrmion motions at RT.** To study the single-skyrmion dynamics, we have applied nanosecond current pulses one by one to the Co$_9$Zn$_9$Mn$_2$-based microdevice and tracked the skyrmion motion at RT. A series of Lorentz transmission electron microscopy (L-TEM) images in Fig. 2 show the single-skyrmion motion driven by sequential 150-ns pulses. The short pulse duration suppresses the skyrmion nucleation/annihilation and the Joule heating effect[4,20]. Upon excitation with an electric current of $j = -6.06 \times 10^{10}$ A m$^{-2}$ flowing from right to left, the circled single skyrmion moves from the lower-left corner to the upper-right corner of the viewing area (Fig. 2a–d, see details in Supplementary Movie 1), exhibiting both the translational displacement ($\triangle x$) and the transverse displacement ($\triangle y$) of the Hall motion at RT (as indicated in Fig. 2e). Under an electric current of $j = 6.32 \times 10^{10}$ A m$^{-2}$ with an opposite direction, i.e., from left to right, both the translational and transverse displacements of the skyrmion are inverted, as shown in Fig. 2f–j (see details in Supplementary Movie 2). The trajectories of the

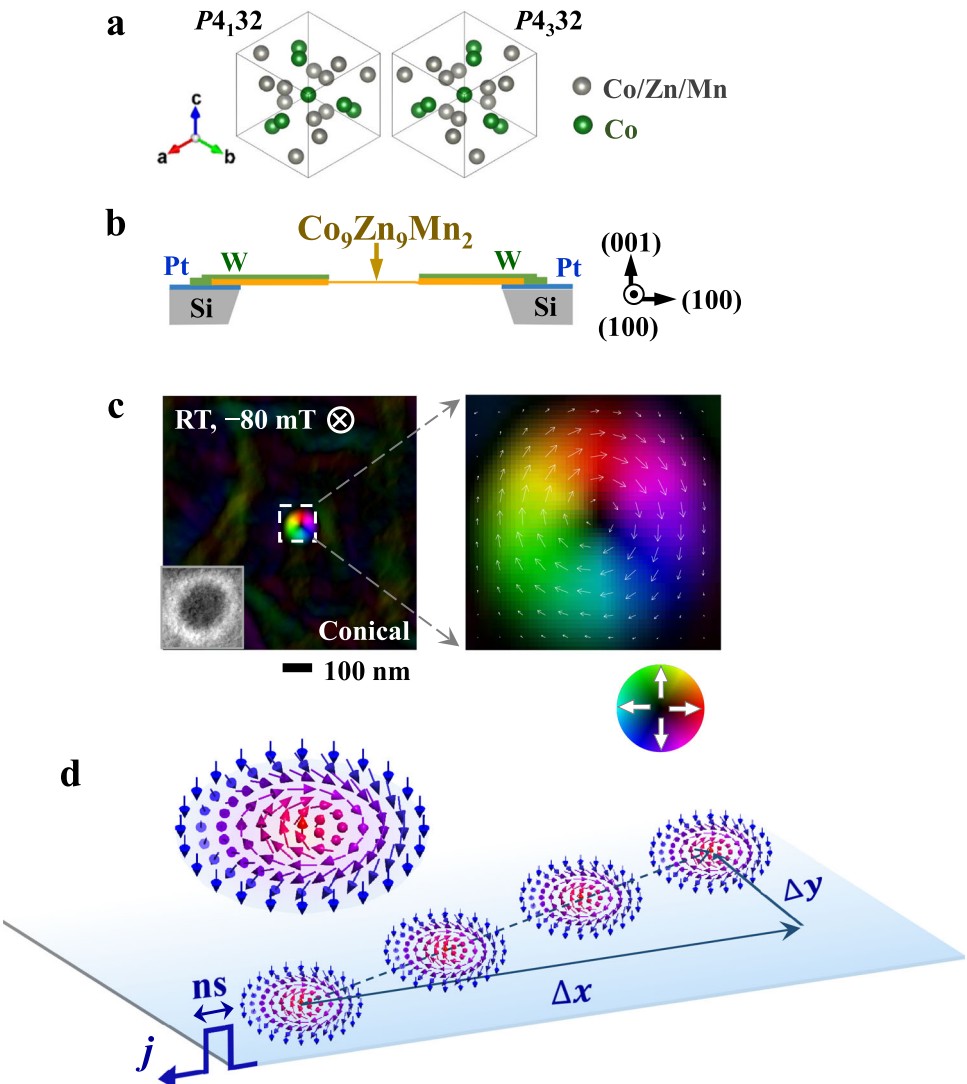

**Fig. 1 A single skyrmion in a Co$_9$Zn$_9$Mn$_2$-based microdevice at room temperature (RT). a** Schematic of the crystal structure of Co$_9$Zn$_9$Mn$_2$ (space group: $P4_132$ or $P4_332$). **b** The cross-section drawing of the microdevice consisting of a (001) Co$_9$Zn$_9$Mn$_2$ thin plate (see details in Supplementary Fig. 1). **c** Magnetic induction maps of a metastable skyrmion generated at RT and a magnetic field of −80 mT applied along the −z-direction. The inset in (**c**) shows the over-focus L-TEM image of the skyrmion. The color wheel encodes the direction of in-plane magnetic components, and dark contrast encodes the out-of-plane components. **d** Schematic of skyrmion motion with the translational displacement (△x) and transverse displacement (△y) induced by pulsed electric current (j) flowing from left to right.

single skyrmion, as marked within the L-TEM images of Fig. 2d, i, demonstrate that the straight skyrmion motions is reversed by reversing the current direction (Fig. 2e, j). The abovementioned L-TEM observations indicate an efficient control of single-skyrmion motion by nanosecond current pulses.

**Reversal of skyrmion Hall motion**. By flipping the direction of the magnetic field applied normally to the thin plate, the topological number is changed from $N_{sk} = +1$ (Fig. 3a) to $N_{sk} = -1$ (Fig. 3f) as defined by Eq. (1). When the electric current with a density j flows along the x-axis, the resultant skyrmion velocities $v_x$ and $v_y$ can be written as[6],

$$v_x = \frac{(4\pi N_{sk})^2 + (\alpha \mathcal{D} + A/v_d)\beta \mathcal{D}}{(4\pi N_{sk})^2 + (\alpha \mathcal{D} + A/v_d)^2} \times \left(-\frac{pa^3}{2eM}j\right) \quad (3)$$

$$v_y = \frac{(\alpha \mathcal{D} - \beta \mathcal{D} + A/v_d)4\pi N_{sk}}{(4\pi N_{sk})^2 + (\alpha \mathcal{D} + A/v_d)^2} \times \left(-\frac{pa^3}{2eM}j\right) \quad (4)$$

where p is the spin polarization, a is the lattice constant, e (>0) is the elementary charge, and A is the pinning term. Note that Eqs. (3) and (4) are based on the mean-field like treatment of the impurity pinning effect represented by the A term, and hence cannot describe the skyrmion creep motion. The $v_y$ is an odd function of $N_{sk}$, and hence the skyrmion Hall motion shows an opposite transverse displacement upon the $N_{sk}$ reversal. On the other hand, since $v_x$ is an even function of $N_{sk}$, the translational skyrmion motion does not change its direction with $N_{sk}$. In Fig. 3b–d, the over-focus L-TEM images show the $N_{sk} = +1$ skyrmion with dark contrast at −80 mT (see details in Supplementary Movie 3). Meanwhile, Fig. 3g–i show the $N_{sk} = -1$ skyrmion with bright contrast in the over-focus L-TEM images at +50 mT with a flipped field direction (see details in Supplementary Movie 4 and Supplementary Fig. 3). The reversed contrast of the skyrmions between Fig. 3b, g indicates the reversed helicity of the skyrmions, in agreement with the expected configurations with fixed DMI and in reversed fields, as shown in Fig. 3a, f. Upon pulsed current stimulation,

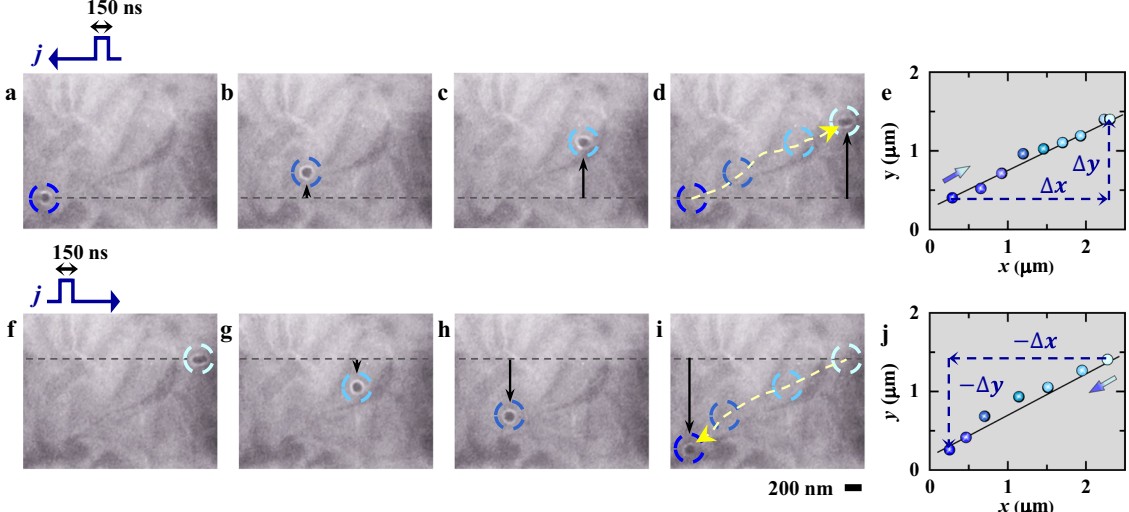

**Fig. 2 Motion tracking of the current-driven single-skyrmion at RT. a–d, f–i** Over-focus L-TEM images showing the single-skyrmion motion stimulated by **a–d** negative ($j = -6.06 \times 10^{10}$ A m$^{-2}$) and **f–i** positive ($j = 6.32 \times 10^{10}$ A m$^{-2}$) pulsed current with a pulse duration of 150 ns at −80 mT and RT. The current flows from right to left for (**a–d**) and from left to right for (**f–i**) as marked by the arrows above (**a**) and (**f**), respectively. The dashed circles mark the skyrmion locations with the dashed arrows in (**d**) and (**i**) showing the directions of skyrmion trajectory. **e, j** Summary of the skyrmion motion along (x-axis) and vertical to (y-axis) the electric current direction: **e** skyrmion positions at $j = -6.06 \times 10^{10}$ A m$^{-2}$ obtained from (**a–d**) and Supplementary Movie 1, and (**j**) those at $j = 6.32 \times 10^{10}$ A m$^{-2}$ obtained from (**f–i**) and Supplementary Movie 2. The straight lines in (**e**) and (**j**) are for the Hall angle estimations, i.e., $\frac{\Delta y}{\Delta x}$ for (**e**) and $\frac{-\Delta y}{-\Delta x}$ for (**j**).

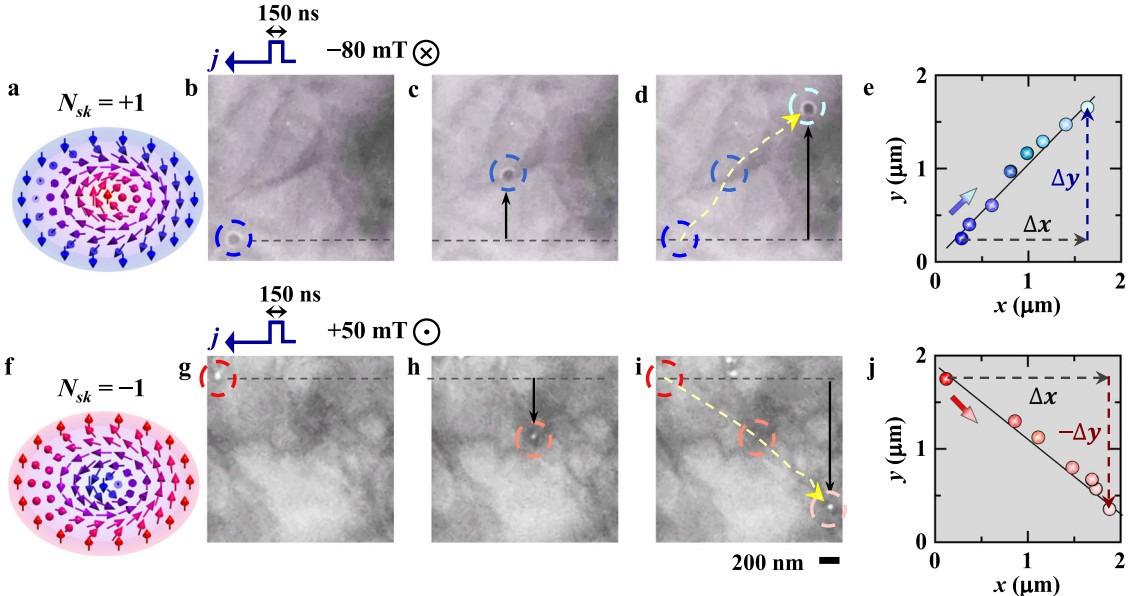

**Fig. 3 Reversal of Hall motion for the single skyrmion with an opposite topological number. a, f** Schematics of skyrmions with the topological number $N_{sk}$ of +1 (**a**) and −1 (**f**). **b–d, g–i** Over-focus L-TEM images showing the reversal of Hall-motion direction at RT: **b–d** for the $N_{sk} = +1$ skyrmion at −80 mT (as denoted by ⊗) and at $j = -5.05 \times 10^{10}$ A m$^{-2}$ and **g–i** for the $N_{sk} = -1$ skyrmion at +50 mT (as denoted by ⊙) and at $j = -4.82 \times 10^{10}$ A m$^{-2}$. The pulsed current flows from right to left as marked by the arrows above (**b**) and (**g**). The dashed circles mark the skyrmion positions with the arrows in (**d**) and (**i**) indicating the direction of the trajectory. **e, j** Summary of the skyrmion traces, with the straight lines showing the translational displacement ($\triangle x$) and the reversed vertical displacements ($\triangle y$ in (**e**) and $-\triangle y$ in (**j**)).

the skyrmion with $N_{sk} = +1$ moves from the lower-left corner to the upper-right corner at $j = -5.05 \times 10^{10}$ A m$^{-2}$ (Fig. 3b–d), while the skyrmion with $N_{sk} = -1$ moves from the upper-left corner to the lower-right corner at $j = -4.82 \times 10^{10}$ A m$^{-2}$ (Fig. 3g–i). Figure 3e, j summarize the locations of $N_{sk} = +1$ and −1 skyrmions, respectively, revealing the same direction of their translational motion and the opposite direction of their Hall motion.

We have measured the skyrmion trace slope to quantify the Hall angle ($\theta_{sk}$) by linearly fitting the skyrmion locations as follows,

$$\theta_{sk} = \tan^{-1} \frac{\triangle y}{\triangle x} \qquad (5)$$

L-TEM observations exemplify the average Hall angle for a pulse duration of 150 ns and a magnetic field of −80 mT; for example,

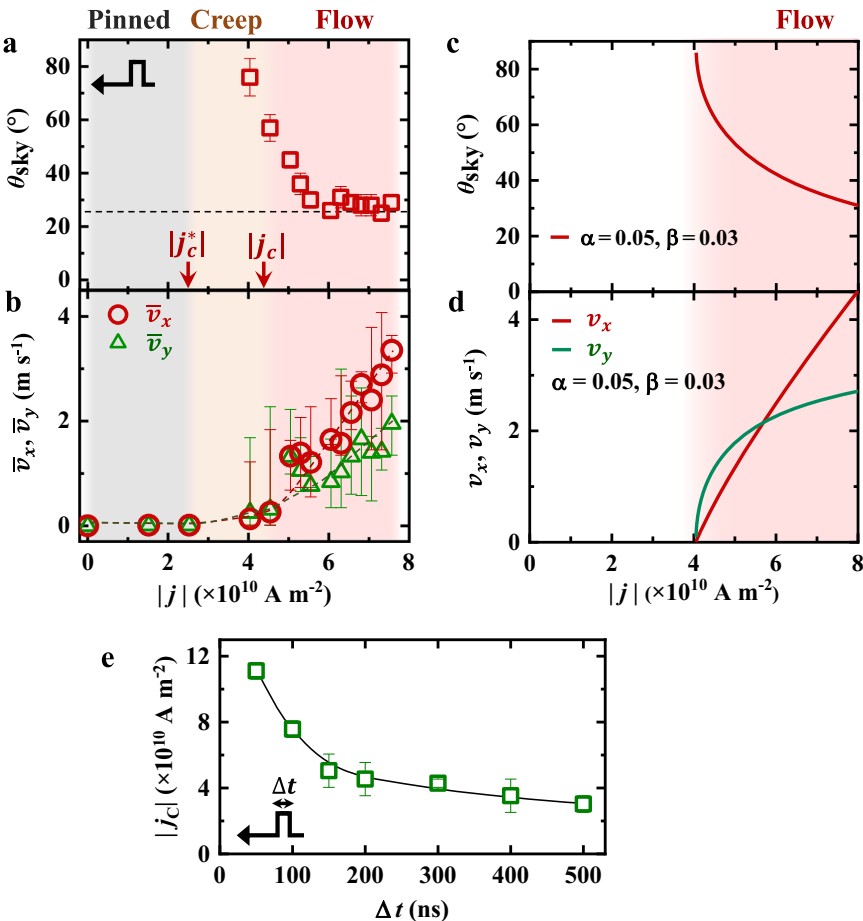

**Fig. 4 Dynamic transition of single-skyrmion motion as a function of pulsed electric current. a**, **b** Evolution of **a** the skyrmion Hall angle ($\theta_{sk}$) and **b** the average skyrmion velocities ($\bar{v}_x$ and $\bar{v}_y$) with electric current density for 150 ns pulse duration and −80 mT magnetic field derived from L-TEM observations. The Hall angle saturates at ~26° with increasing electric current, as marked by the dashed line in (**a**). $\bar{v}_x$ (red color) and $\bar{v}_y$ (green color) in (**b**) are the average transverse and longitudinal velocities of the single skyrmion, respectively. The error bars in (**b**) show the maximum and minimum velocities for a single current pulse. The gray, orange, and pink regions in (**a**, **b**) correspond to the skyrmion-pinned regime ($|j| < |j_C^*|$ ~2.52 × 10¹⁰ A m⁻²), the skyrmion creep-motion regime (2.52 × 10¹⁰ A m⁻² < $|j|$ < 4.54 × 10¹⁰ A m⁻²), and the skyrmion linear flow-motion regime ($|j| > |j_C|$ ~4.54 × 10¹⁰ A m⁻²), respectively. **c**, **d** Calculations of $|j|$-dependent (**c**) Hall angle $\theta_{sk}$ and (**d**) velocities of $v_x$ (red line) and $v_y$ (green line) with $\alpha = 0.05$ and $\beta = 0.03$ for the flow regime of skyrmion motion. **e** A plot of the critical current density ($|j_C|$), driving a single-skyrmion motion into the flow regime, as a function of the pulse duration $\triangle t$.

$\theta_{sk}$ is ~26° at $j = -6.06 \times 10^{10}$ A m⁻² (Fig. 2a–e), while $\theta_{sk}$ is increased to ~45° with reducing the magnitude of electric current to $j = -5.05 \times 10^{10}$ A m⁻² (Fig. 3a–e). L-TEM images of the single-skyrmion motion at various current densities are presented in Supplementary Fig. 4.

**Dynamic transition of the single-skyrmion motion.** Figure 4a, b summarize the $|j|$-dependent skyrmion Hall angle and velocity with the current flowing from left to right. The average values of $\bar{v}_x$ and $\bar{v}_y$ are estimated from the displacement of the skyrmion after $n$ pulses with the duration of $\triangle t = 150$ ns as follows,

$$\bar{v}_x = \frac{\triangle x}{n \cdot \triangle t} \qquad (6)$$

$$\bar{v}_y = \frac{\triangle y}{n \cdot \triangle t} \qquad (7)$$

The error bars in Fig. 4b show the maximum and minimum skyrmion velocities for a single-pulse current stimulation. Previous studies[11,12,27] predict a low depinning force ($j_C^*$) for a

creep motion and a relatively high critical electric current ($j_C$) for a flow motion of skyrmions. Experimentally when we apply a relatively small pulsed current with the magnitude below $2.52 \times 10^{10}$ A m⁻², i.e., ($|j| < |j_c^*|$) (as marked in Fig. 4a), the skyrmion does not move, perhaps due to the presence of pinning potential, hence exhibiting a zero velocity (the gray region in Fig. 4a, b). When slightly increasing the current, i.e., for $|j_c^*|$ ($2.52 \times 10^{10}$ A m⁻²) < $|j|$ < $|j_c|$ ($4.54 \times 10^{10}$ A m⁻²), the onset of skyrmion movement is observed: The skyrmion shows a creep motion under sequential current pulses (the orange region in Fig. 4a, b), which leads to a relatively small average velocity below 0.24 m s⁻¹. Meanwhile, the $\bar{v}_y$ is larger than $\bar{v}_x$, i.e., $\frac{\bar{v}_y}{\bar{v}_x} > 1$, hence the calculated Hall angle is rather large at ~76° for $|j|$ just above $|j_c^*|$. When $|j|$ is further increased over $|j_c|$ ($4.54 \times 10^{10}$ A m⁻²), the skyrmion exhibits a linear flow motion (the pink region in Fig. 4a, b), as demonstrated in Figs. 2, 3 and Supplementary Figs. 3, 4, and Supplementary Movies 1–4. With increasing current density, the skyrmion dynamics change from the creep motion to the flow motion, and the Hall angle decreases monotonically and eventually saturates at ~26° (Fig. 4a). On the other hand, the velocity of the

single skyrmion increases with the electric current (Fig. 4b). The $\bar{v}_x$ increases slightly faster than $\bar{v}_y$, leading to $\frac{\bar{v}_y}{\bar{v}_x} < 1$ at $|j| > |j_c|$. The maximum velocities $\bar{v}_x$ and $\bar{v}_y$ reach ~3.34 m s$^{-1}$ and ~1.95 m s$^{-1}$, respectively, at $|j|$ ~7.57 × 10$^{10}$ A m$^{-2}$.

Figure 4c, d show plots of $|j|$ versus $\theta_{sk}$, $v_x$ and $v_y$ using Eqs. (3) and (4) at $\alpha = 0.05$ and $\beta = 0.03$ (see details in Methods). The fitting curves represent the skyrmion dynamics in a flow regime. As $|j|$ increases, $v_x$ (red line in Fig. 4d) and $v_y$ (green line in Fig. 4d) increase monotonically, and the $\theta_{sk}$ (Fig. 4c) decreases and tends to approach a saturated value. As $|j|$ decreases, the skyrmion velocity decreases, and the calculated $\theta_{sk}$ shows a rapid increase due to the impurity pinning effect, as the skyrmion is deflected by the Magnus force when the skyrmion approaches defects such as a grain boundary[9,11]. The calculated profiles of $|j| - \theta_{sk}$, $|j| - v_x$, and $|j| - v_y$ (Fig. 4c, d) coincide with our present in situ L-TEM observations for single-skyrmion dynamics in the flow-motion regime.

Let us move onto the pulse-duration dependence of the skyrmion flow motion (Fig. 4e). At a large $\triangle t$ of 500 ns, an electric current density above $|j_c|$ of 3.03 × 10$^{10}$ A m$^{-2}$ can induce the skyrmion flow motion; this density is two orders of magnitude smaller than that needed for driving ferromagnetic domain walls[28]. As $\triangle t$ is reduced to 50 ns, the $|j_c|$ is increased up to 1.1 × 10$^{11}$ A m$^{-2}$. A large $\triangle t$ above ~1 ms, as previously discussed for the FeGe helimagnet[20], will result in a significant Joule heating effect. The short pulse duration of the nanosecond used in this work reduces the Joule heating and hence promotes the STT effect for the current-driven skyrmion dynamics.

## Discussion

Our systematic measurements demonstrate the single-skyrmion motion as a function of electric current at RT in the chiral-lattice magnet Co$_9$Zn$_9$Mn$_2$. Under the pulsed current stimulation, the skyrmion exhibits a trapping-limited motion: it moves during the short current pulse and then is trapped by nearby pinning sites after the stimulus, where the skyrmion can deform, as exemplified by Fig. 2c, d and Supplementary Fig. 4m. The L-TEM captures the two static skyrmion states before/after the current pulse, while the intermediate state during the pulse could not be detected with our experimental setup, suggesting that the skyrmion velocity may be larger than measured values. The measured $\bar{v}_x$ reaches ~3.34 m s$^{-1}$ at $|j| = 7.57 \times 10^{10}$ A m$^{-2}$ (Fig. 4b), nearly consistent with our calculations (Fig. 4d). The Hall angle is rather large in the creep-motion regime since the skyrmion is easily trapped by pinning sites at a small $|j|$ just above $|j_C^*|$. In contrast, at $|j| > |j_C|$, the skyrmion motion depends less on the pinning sites, demonstrating a saturated Hall angle of ~26° in Co$_9$Zn$_9$Mn$_2$, which is determined by material parameters[11,13,20]. The plot of $|j|$-$\theta_{sk}$ (Fig. 4a) observed experimentally fits well with our calculations (Fig. 4c) for the STT-induced Bloch-type skyrmion motion in chiral-lattice magnets; in fact, the skyrmion dynamics observed here are in contrast to the SOT-induced Néel-type skyrmion motion in heterostructured thin films, where the $\theta_{sk}$ is close to zero for a small $j$ and exhibits a monotonic increase with increasing the driving force[13]. The mechanism of the spin-polarized electric current acting on skyrmions, i.e., the STT in chiral-lattice magnets or the SOT in heterostructured thin films, may determine the skyrmion dynamic behavior in the system with impurities.

In summary, we have experimentally revealed the transition of skyrmion dynamics from the pinned state to the flow motion via a creep motion as a function of electric current in Co$_9$Zn$_9$Mn$_2$, and directly demonstrated the reversal of skyrmion Hall motion with an opposite topological number. Our achievements provide evidence of STT-driven skyrmion dynamics upon nanosecond-pulse current

excitation in chiral-lattice magnets, deepen the understanding of the pinning effect on skyrmion dynamics, and will promote the studies of dynamical motions of various topological spin textures.

## Methods

**Microdevice preparation**. Single crystals of Co$_9$Zn$_9$Mn$_2$ were grown by the Bridgman method and examined by X-ray diffraction. The microdevice was prepared using a focused ion beam (FIB) system equipped with a gallium ion gun (NB-5000, Hitachi, Japan). It consisted of a (001) Co$_9$Zn$_9$Mn$_2$ thin plate whose left and right edges were connected to two Pt leads using tungsten (W) layers. The top and bottom edges were additionally padded by amorphous carbon (see Supplementary Fig. 1). The orientation of the Co$_9$Zn$_9$Mn$_2$ thin plate was checked via a selected area diffraction pattern.

**L-TEM observations**. Real-space observations of skyrmions were performed using JEM-2100F and JEM-2800F microscopes (JEOL, Japan) at an acceleration voltage of 200 kV. A magnetic field was applied normally to the thin plate, and its magnitude was controlled by changing the objective-lens current of the TEM. The L-TEM images under the magnetic field applied along the +z-direction were recorded with the JEM-2800F microscope, and the field strength was measured by a Hall probe. The L-TEM images under the magnetic field applied along the −z-direction were recorded in the JEM-2100F microscope. The value of nanosecond pulsed current was calculated by dividing the pulsed voltage by sample resistance and sample cross-section area. The pulsed voltage was applied by an arbitrary function generator (AFG 31252, Tektronix, U.S.). Quantitative in-plane induction field maps were obtained by analyzing the under- and over-focus L-TEM images using the software package QPt (HREM Co., Japan) based on the transport-of-intensity equation (TIE)[29].

**Calculations**. The calculations of the skyrmion Hall angle and velocity (Fig. 4c, d) are based on the Thiele Eq. (2) in the presence of impurities as given in ref. [6]. The drift velocity of a skyrmion $v_d$ is given by

$$v_d = \frac{\sqrt{(\mathcal{D}\alpha A)^2 + ((4\pi N_{sk})^2 + (\mathcal{D}\alpha)^2)((\beta^2 \mathcal{D}^2 + (4\pi N_{sk})^2)v_s^2 - A^2)}}{(4\pi N_{sk})^2 + (\mathcal{D}\alpha)^2} \quad (8)$$

where $v_s = -\frac{pa^3}{2em}j$ is the velocity of the conduction electrons. The parameters are given as follows[6,30]: dissipative force $\mathcal{D} = 5.577\pi$, damping factor $\alpha = 0.05$, non-adiabatic coefficient $\beta = 0.03$, pinning term $A \equiv 4\pi v_{pin}$ with the pinning velocity $v_{pin} = 3.2$, lattice constant $a = 6.32$ A, local magnetic moment $M = 1$, and spin polarization $p = 0.1$ which was estimated from the ratio of magnetizations in the skyrmion phase to the saturated magnetization at 2 K[24,30].

## Data availability

The data that support the findings of this study are available from the corresponding author upon reasonable request.

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

## Acknowledgements
We thank Konstantin V. Iakoubovskii, Wataru Koshibae, Jan Masell, Fumitaka Kagawa, Fehmi S. Yasin, and Mari Ishida for enlightening discussions. We thank Tomoka Kikitsu and Daisuke Hashizume (Materials Characterization Support Team in the RIKEN Center for Emergent Matter Science) and RIKEN CEMS Emergent Matter Science Research Support Team for technical supports with TEM (JEM-2100F) and FIB (NB-5000, Hitachi), respectively. This work was partly supported by Grants-In-Aid for Scientific Research (A) (Grant No.19H00660) from JSPS, and Japan Science and Technology Agency CREST (Grant No. JPMJCR1874 and No. JPMJCR20T1) from JST.

## Author contributions
L.P., Y.Taguchi, Y.Tokura, and X.Y. jointly conceived the project. L.P. designed and fabricated the microdevice, performed the L-TEM observations, and analyzed the experimental results with X.Y. L.P. performed the calculations with N.N. K.K. synthesized the $Co_9Zn_9Mn_2$ crystal with Y.Taguchi. L.P. wrote the manuscript with inputs and comments from all authors.

## Competing interests
The authors declare no competing interests.
