## [Peer Review File · Nature Communications]

Reviewers' Comments:

Reviewer #1:

None

Reviewer #2:

Remarks to the Author:

Comments: The present work focuses on the systematic measurements of the single-skyrmion motion under the electric current at the room temperature, clearly revealing the dynamic transition from the pinned-static state to the linear flow motion via a creep event, in accord with the theoretical prediction. Although there are some skyrmion studies in this series of CoZnMn alloy since the initial work from this group. I would still categorize this work as a new and innovative approach on the subject of electric skyrmion motion studies, different enough from previous studies. The experimental observation is interesting and the experimental result is solid. I am inclined to recommend for publication once the authors satisfactorily address my questions and comments below.

Question 1. The skyrmion contrast of LTEM images in the Figure 3 of Co₈Zn₈Mn₄ (Nano Lett. 2017, 17, 3, 1637–1641) has a similar additional contrast circle with what you observed, which is clearly different from the LTEM images in the Figure 2,3 of Co₈Zn₉Mn₃ and other DMI skyrmion like FeGe. Can you tell the differences about the skyrmion number, spin configuration and the electric driven abilities for these two different domain contrasts? And what is the generation reason for this kind of spin configuration.

Question 2. It is better to give the corresponding LTEM image showing the domain contrast together with TIE spin configuration in the Figure 1. The additional circular contrast at the outside skyrmion can be clearly seen in the Figure 2 but can't be seen in the TIE image?

Question 3. It is better to make a statement why you create a metastable SkL by a pulsed current at -40 mT not by just magnetic field. Is it a requirement for this electric driven experiment?

Reply to Reviewer #2

The present work focuses on the systematic measurements of the single-skyrmion motion under the electric current at the room temperature, clearly revealing the dynamic transition from the pinned-static state to the linear flow motion via a creep event, in accord with the theoretical prediction. Although there are some skyrmion studies in this series of CoZnMn alloy since the initial work from this group. I would still categorize this work as a new and innovative approach on the subject of electric skyrmion motion studies, different enough from previous studies. The experimental observation is interesting and the experimental result is solid. I am inclined to recommend for publication once the authors satisfactorily address my questions and comments below.

We thank the Reviewer for his/her positive comments and recommendation for publication of our paper in *Nature Communications*.

1. The skyrmion contrast of LTEM images in the Figure 3 of Co₈Zn₈Mn₄ (Nano Lett. 2017, 17, 3, 1637–1641) has a similar additional contrast circle with what you observed, which is clearly different from the LTEM images in the Figure 2,3 of Co₈Zn₉Mn₃ and other DMI skyrmion like FeGe. Can you tell the differences about the skyrmion number, spin configuration and the electric driven abilities for these two different domain contrasts? And what is the generation reason for this kind of spin configuration.

Reply: We thank the Reviewer for carefully reading. The additional bright L-TEM contrast outside the skyrmion emerges because the electron beam is divergently deflected from the center to the periphery of the skyrmion, thus resulting in a dark core and a bright rim in L-TEM images. It is commonly observed by L-TEM for thermodynamic equilibrium skyrmions in a lattice state (e.g. in Co₈Zn₈Mn₄, [Morikawa *et al*, *Nano Lett.* 2017, 17, 3, 1637–1641], Co₈Zn₉Mn₃, [Tokunaga *et al*, *Nat. Commun.* 2015, 6, 7638; Yu *et al*, *Nature* 2018, 564, 95-98], and FeGe, [Yu *et al*, *Nat. Phys.* 2018, 14, 832-836]). This bright circular contrast is not induced by any additional magnetic texture and hence does not change the topological number of the skyrmion, which has been confirmed by the transpot-of-intensity (TIE) analyses and current-driven skyrmion Hall motions.

We agree with the Reviewer that the current-driven dynamics of spin textures rely on their topological number N_{sk} . The skyrmion with $N_{sk} = +1$ or -1 allows the skyrmion Hall motion, which is transverse to the current direction, as shown in the present work. While the skyrmionium (possibly concerned by the

Reviewer) observed in FeGe (Ref. [Tang et al. *Nat. Nanotech.* <https://doi.org/10.1038/s41565-021-00954-9> (2021)]), showing a different L-TEM contrast with that for skyrmions, has a zero topological number. Thus, only a translational motion (no Hall motion) of such skyrmionium appears upon application of electric current.

Apart from the thermodynamic equilibrium skyrmion phase near T_C , fertile exotic spin textures have already been excited as metastable states with external stimuli, such as the metastable skyrmions [Yu et al, *Nat. Phys.* 2018, 14, 832-836], skyrmionium [Tang et al, *Nat. Nanotech.* <https://doi.org/10.1038/s41565-021-00954-9> (2021)], domain wall bimeron [Nagase et al, *Nat. Commun.* 2021, 12, 3490], and meron-antimeron pair [Yu et al, *Nature* 2018, 564, 95-98]. Phenomenologically, the electric current is more efficient in creating various metastable topological objects than the external magnetic field due to the spin-transfer torque effect induced by the spin-polarized current.

2. It is better to give the corresponding LTEM image showing the domain contrast together with TIE spin configuration in the Figure 1. The additional circular contrast at the outside skyrmion can be clearly seen in the Figure 2 but can't be seen in the TIE image?

Reply: We thank the Reviewer for this suggestion. We have inserted an L-TEM image of the skyrmion in the revised Fig. 1 (see below Fig. R1).

As explained above in our reply to comment #1, the bright contrast outside the skyrmion is reasonable in the defocused L-TEM image because the electron beam is divergently deflected from the center to the periphery of the skyrmion, thus resulting in a dark core and a bright rim in L-TEM images. This bright rim is not induced by any additional magnetic texture, which is directly confirmed by the corresponding field map obtained by TIE analyses, as shown in Fig. R1c.

Fig. R1. A single skyrmion observed in a $\text{Co}_9\text{Zn}_9\text{Mn}_2$ -based microdevice at room temperature (RT). (a) Schematic of the crystal structure of $\text{Co}_9\text{Zn}_9\text{Mn}_2$ (space group: $P4_132$ or $P4_332$). (b) The cross-section drawing of the microdevice consisting of a (001) $\text{Co}_9\text{Zn}_9\text{Mn}_2$ thin plate (see details in Supplementary Fig. 1). (c) Magnetic induction maps of a metastable skyrmion generated at RT and a magnetic field of -80 mT applied along the $-z$ -direction. **The inset in (c) shows the over-focus L-TEM image of the skyrmion.** The color wheel encodes the direction of in-plane magnetic components, and dark contrast encodes the out-of-plane components. (d) Schematic of skyrmion motion with the translational displacement (Δx) and transverse displacement (Δy) induced by pulsed electric current (j) flowing from left to right.

3. It is better to make a statement why you create a metastable SkL by a pulsed current at -40 mT not by just magnetic field. Is it a requirement for this electric driven experiment?

Reply: We thank the Reviewer for enlightening this point. We have added the following statement to explain why do we use pulsed current to create the SkL in the revised manuscript (page 4, lines 77-79):

“The application of magnetic field will only generate a conical state, while the SkL is suppressed at RT²⁴. Therefore we use electric current pulses to create the metastable SkL in the thermodynamically stable conical phase at RT (Supplementary Fig. 2).”

Let us elaborate some details. The skyrmion lattice (SkL) only occupies a narrow region in the temperature-magnetic field phase diagram, just below $T_C \sim 396$ K, while the conical state is thermodynamically stable at room temperature (RT) under the application of magnetic field [Karube *et al. Phys. Rev. B* 2020, 102, 064408; Yu *et al. Nano lett.* 2020, 20, 7313-7320]. To enable the SkL to persist at RT, we applied pulsed current together with a weak magnetic field to nucleate the metastable skyrmions. This procedure is required for the creation of metastable skyrmions and hence the study of their current-driven dynamics.

List of changes

1. Modification of Fig. 1: we have added an L-TEM image of the skyrmion in the inset of Fig. 1c.
2. On page 4, lines 77-79, we added a statement to explain why do we use pulsed current to create the skyrmion lattice (SkL), as follows,
“The application of magnetic field will only generate a conical state, while the SkL is suppressed at RT²⁴. Therefore we use electric current pulses to create the metastable SkL in the thermodynamically stable conical phase at RT (Supplementary Fig. 2).”
3. Modification of the article format to fix the style of *Nature Communications*: we have added subheadings for the Results section.

Reviewers' Comments:

Reviewer #2:

Remarks to the Author:

My comments have been appropriately addressed in the response letter. I have no further questions and recommend to publish the work in Nature Communication.

Comments from Reviewer #1

The authors demonstrate the motion and control of a single-skyrmion in a chiral magnet using nano-second duration current pulses. They observe the expected inversion of the Hall motion when they change the sign of the skyrmion number. Their experimental study also reveals the dynamic transition from the pinned static state to the linear flow motion via an intermediate creep regime, in agreement with theoretical predictions.

While the theory of this phenomenon is very well-known by now, the experimental results presented in this work are very valuable because they demonstrate a degree of control of single skyrmion motion at room temperature, which confirms that these systems are indeed promising for future applications to spintronics. The good agreement between theory and experiment in the flow regime also confirms that the phenomenon is well understood. The videos are very illustrative, and the overall presentation of the experimental results is outstanding. I believe that this manuscript will trigger more experimental studies of the skyrmion dynamics induced by pulsed currents. For instance, it is known that a large-enough currents can produce other dynamical transitions, such as the spontaneous generation of new skyrmions.

I therefore believe that this manuscript deserves to be published in *Nature Communications*.

Reply: We thank the Reviewer for his/her high evaluation of our paper and recommendation for publication in *Nature Communications*.

I just recommend the authors to improve the presentation of equations (3) and (4). My basic objection is that these equations are obtained by applying a mean field approximation to the pinning force. That approximation, which is only valid in the flow regime, leads to the parameter A that appears in both equations. In the current version of the manuscript, the meaning of the parameter A , the nature of the approximation that leads to equations (3) and (4), and the regime of validity of these equations do not become clear until the end of the paper (more specifically line 159). This is not the most pedagogical way of introducing the theoretical results. In my view, this discussion should appear right next to equations (3) and (4), so the reader can immediately understand the origin, meaning and limitations of these

equations.

Reply: We thank the Reviewer for the suggestion. We have added the following discussion (as highlighted in red) in the revised manuscript (page 6, lines 113-114),

“When the electric current with a density j flows along the x -axis, the resultant skyrmion velocities v_x and v_y can be written as⁶,

$$v_x = \frac{(4\pi N_{sk})^2 + \left(\alpha\mathcal{D} + \frac{A}{v_d}\right)\beta\mathcal{D}}{(4\pi N_{sk})^2 + (\alpha\mathcal{D} + A/v_d)^2} \times \left(-\frac{pa^3}{2eM}j\right) \quad (3)$$

$$v_y = \frac{\left(\alpha\mathcal{D} - \beta\mathcal{D} + \frac{A}{v_d}\right)4\pi N_{sk}}{(4\pi N_{sk})^2 + (\alpha\mathcal{D} + A/v_d)^2} \times \left(-\frac{pa^3}{2eM}j\right) \quad (4)$$

where p is the spin polarization, a is the lattice constant, e (>0) is the elementary charge, and A is the pinning term. **Note that eqs.(3) and (4) are based on the mean-field like treatment of the impurity pinning effect represented by the A term, and hence can not describe the skyrmion creep motion.”**

Comment from Reviewer #2

My comments have been appropriately addressed in the response letter. I have no further questions and recommend to publish the work in Nature Communication.

Reply: We thank that the Reviewer finds our revised manuscript and responses satisfactory, and recommend publication of our paper in *Nature Communications*.